# Integrative Analysis of Transcriptome and Metabolome Reveals the Pivotal Role of the NAM Family Genes in *Oncidium hybridum* Lodd. Pseudobulb Growth

**DOI:** 10.3390/ijms251910355

**Published:** 2024-09-26

**Authors:** Yi Liu, Qing Zhu, Zukai Wang, Haoyue Zheng, Xinyi Zheng, Peng Ling, Minqiang Tang

**Affiliations:** Key Laboratory of Genetics and Germplasm Innovation of Tropical Special Forest Trees and Ornamental Plants (Ministry of Education), Collaborative Innovation Center, School of Tropical Agriculture and Forestry, Hainan University, Haikou 570228, China; 22220954000016@hainanu.edu.cn (Y.L.); 22210907000001@hainanu.edu.cn (Q.Z.); 21220954000037@hainanu.edu.cn (Z.W.); 21220954000049@hainanu.edu.cn (H.Z.); 23220954000032@hainanu.edu.cn (X.Z.)

**Keywords:** NAM gene family, *Oncidium hybridum*, transcriptome analysis, metabolome analysis, pseudobulb growth

## Abstract

*Oncidium hybridum* Lodd. is an important ornamental flower that is used as both a cut flower and a potted plant around the world; additionally, its pseudobulbs serve as essential carriers for floral organs and flower development. The NAM gene family is crucial for managing responses to various stresses as well as regulating growth in plants. However, the mechanisms by which NAM genes regulate the development of pseudobulbs remain unclear. In this study, a total of 144 NAM genes harboring complete structural domains were identified in *O. hybridum*. The 144 NAM genes were systematically classified into 14 distinct subfamilies via phylogenetic analysis. Delving deeper into the conserved motifs revealed that motifs 1–6 exhibited remarkable conservation, while motifs 7–10 presented in a few NAM genes only. Notably, NAM genes sharing identical specific motifs were classified into the same subfamily, indicating functional relatedness. Furthermore, the examination of occurrences of gene duplication indicated that the NAM genes display 16 pairs of tandem duplications along with five pairs of segmental duplications, suggesting their role in genetic diversity and potential adaptive evolution. By conducting a correlation analysis integrating transcriptomics and metabolomics at four stages of pseudobulb development, we found that *OhNAM023*, *OhNAM030*, *OhNAM007*, *OhNAM019*, *OhNAM083*, *OhNAM047*, *OhNAM089*, and *OhNAM025* exhibited significant relationships with the endogenous plant hormones jasmonates (JAs), hinting at their potential involvement in hormonal signaling. Additionally, *OhNAM089*, *OhNAM025*, *OhNAM119*, *OhNAM055*, and *OhNAM136* showed strong links with abscisic acid (ABA) and abscisic acid glucose ester (ABA-GE), suggesting the possible regulatory function of these NAM genes in plant growth and stress responses. The 144 NAM genes identified in this study provide a basis for subsequent research and contribute to elucidating the intricate molecular mechanisms of NAM genes in Oncidium and potentially in other species.

## 1. Introduction

*Oncidium hybridum* Lodd., an important plant in the Orchidaceae family, is famous for its unique flower shape and rich array of colors. *O. hybridum* has high ornamental value and is one of the main commercial orchids; it is primarily cultivated as a potted plant and as cut flowers. Pseudobulbs are specialized stems unique to orchids, which are capable of producing both leaf and flower buds while storing water and nutrients. The growth and decline of pseudobulbs exert a direct influence on the development of orchids, especially the formation of floral organs [1]. Since orchids are ornamental plants, most existing research has focused on the flowers, with limited studies dedicated to examining the regulatory mechanisms involved in pseudobulb growth. Wang et al. discovered that *DOFT* promotes flowering in *Dendrobium* Chao Praya Smile. Interestingly, half of the 35S:DOFT transgenic orchids exhibited pseudobulb formation 8 to 10 weeks earlier than the non-transgenic orchids, indicating that *DOFT* also plays a role in promoting pseudobulb formation [2]. Li et al. found that *DOTFL1* has a negative impact on pseudobulb formation and flowering in *Dendrobium* orchids and speculated that *DOTFL1* disrupts normal floral organ development by inhibiting pseudobulb formation [3]. Presently, there is an increasing demand for *O. hybridum* in the flower market, leading to an expansion in the scale of production. However, the growth mechanism of *O. hybridum* pseudobulbs remains unclear, which constrains the further development of the *O. hybridum* industry. Therefore, studying the growth and development of pseudobulbs holds significant practical value for production. *O. hybridum* also serves as an ideal model organism for research on the Orchidaceae family. Consequently, investigating the growth mechanism of pseudobulbs holds significant theoretical value and will play a pivotal role in advancing research on Orchidaceae plants.

Transcription factors (TFs) are protein molecules that possess specific structures in eukaryotes, serving as regulators of gene transcription [4]. According to DNA structural domains, plant TFs can be classified into several families, including NAC, WRKY, ARF, DOF, and MYC. The NAC gene family is among the most thoroughly researched TFs, and it is widespread in land plants. To date, the Plant Transcription database (https://planttfdb.gao-lab.org/, accessed on 17 March 2024) has collected NAC genes from 166 species [5]. The abbreviation of the NAC gene family originates from the initials of NAM, ATAF1/2, and CUC2 [6]. At the N-terminus of the NAC gene is a conserved DNA-binding domain that normally consists of 150–160 amino acid residues. Additionally, it features a flexible transcriptional regulatory region at the C-terminus [7]. Some NAC proteins contain a NAM motif in the DNA-binding domain. The NAM domain is highly conserved, and it can be categorized into five subdomains (A to E). Previous research had shown that, compared to CUC2 and ATAF1/2 proteins in the NAC family, NAM proteins possess many distinctive features. It was reported that the NAM proteins contain a large number of basic amino acids (histidine, arginine, and lysine), and that the positive and negative amino acids are unevenly distributed across each subdomain [8]. In addition, some research has found that subdomain B is rich in acidic amino acids, whereas subdomains C and D are deficient in acidic amino acids but abundant in basic amino acids [9].

Previous studies indicate that the NAM genes are crucial for responding to various stresses and are also vital to plant growth. Uauy et al. discovered that reducing the RNA levels of multiple homologs of NAM through RNA interference can delay wheat senescence [10]. You J. et al. discovered that *OsPP18*, regulated by the NAM gene *SNAC1*, confers oxidative stress and drought resistance in rice [11]. Other researchers conducted an integrative analysis of transcriptome and metabolome, further elucidating how the NAM gene family controls plant growth by modulating endogenous hormones. The concentration of auxin (IAA) regulated by *PIN1* determines the formation and distribution of leaf margin serrations, while the polarity of PIN1 localization is controlled by a class of TFs, specifically, NAM/CUC [12]. The overexpression of *ClNAM* in *Chrysanthemum lavandulifolium* (Fisch. ex Trautv.) Makino leads to the incomplete development of ray florets; meanwhile, inhibiting the expression of *ClNAM* led chrysanthemum callus tissue to differentiate into adventitious buds that ceased growth after initiation. Further analysis suggests that *ClNAM* might regulate the transport of IAA by modulating the expression levels of *ClPINs*, thereby regulating the process of flower differentiation and development [13]. ABA contributes to the postponement of tuber dormancy in *Gladiolus hybridus* Hort.; Wu J. et al. discovered that *GhNAC83* negatively regulates an ABA signal-regulating protein, *GhPP2C1*. *GhNAC83* inhibits the expression of *GhPP2C1* and the cytokinin biosynthesis gene *GhIPT* by binding to their promoter regions. This action promotes ABA biosynthesis and suppresses cytokinin biosynthesis, ultimately delaying the release of tuber dormancy in *G. hybridus* [14]. However, the identification of NAM genes and their mechanisms of regulation in relation to endogenous plant hormones have not been reported in *O. hybridum* pseudobulbs.

In this study, we identified the NAM genes in *O. hybridum* using bioinformatics methods. Their physicochemical properties, chromosomal location, gene duplication, gene structure, conserved motifs, and phylogenetic relationships were systematically investigated. Furthermore, by conducting a combined analysis of transcriptomics and metabolomics, we delved deeper into the intricate molecular mechanisms underlying the NAM gene family’s regulation of the growth and development of pseudobulbs. Overall, this study lays the foundation for future research focused on validating the functional roles of NAM genes in *O. hybridum*.

## 2. Results

### 2.1. Identification and Physicochemical Property Analysis of NAM Genes

A total of 144 NAM genes were successfully identified using two methods, after filtering out sequences that lacked the NAM domain. To assign each NAM gene an abbreviation, we adopted a systematic naming pattern. First, all NAM gene names commenced with a species-specific term (Oh). Second, we inserted the gene family acronym (NAM) in the middle of each name. Finally, each gene name concluded with a sequential number corresponding to the gene’s position on the chromosomes. Following this systematic naming pattern, these genes were sequentially designated as *OhNAM001* to *OhNAM144*. The sequence information of 144 NAM proteins is provided in Appendix A.

The physicochemical characteristics of the NAM proteins were examined utilizing TBtools; detailed information is provided in Appendix A. The results revealed that the amino acids of these NAM proteins ranged from 98 (*OhNAM031*) to 773 (*OhNAM004*) amino acids, the molecular weight ranged from 11.42 (*OhNAM031*) to 86.46 (*OhNAM004*) kDa, and the isoelectric point (pI) values ranged from 4.49 (*OhNAM038*) to 10.1 (*OhNAM104*). The subcellular localization analysis results obtained using WoLF PSORT and Plant-mPLoc both indicate that most of the NAM genes are nuclear proteins, and only *OhNAM013*, *OhNAM101*, *OhNAM102*, and *OhNAM112* are localized in both the chloroplast and nucleus.

### 2.2. Chromosomal Location, Duplication Analysis, and Collinearity with Arabidopsis thaliana (L.) Heynh

The position information of the NAM genes was derived from the gff file, revealing that these genes are unevenly spread across 39 chromosomes and scaffolds (Figure 1). The results revealed that most of NAM genes were distributed on Chr 5 (13 genes; 9.0%), followed by Chr 4 (9 genes; 6.3%). Chr 8, 11, 14, 18, 19, 25, 26, 31, 36, and 40 contained the fewest NAM genes, with only one present for each chromosome. The quantity of genes found on different chromosomes varied between 2 and 6 NAM genes. Notably, Chr 22, 32, and 37 did not contain any NAM genes.

We utilized MCScanx (version 1) to investigate tandem duplication (TDs) events in the NAM genes. The results revealed that, among the 39 chromosomes, 16 pairs of tandem duplicate genes were discovered on nine chromosomes (Chr 4, 5, 10, 15, 17, 29, 30, 38, and 39). To obtain evolutionary information about these NAM genes, the Ka/Ks values for the duplicated gene pairs were calculated. Remarkably, every gene pair had a Ka/Ks value of less than 1, suggesting that negative selection played a role in their evolution. However, we were unable to calculate the Ka/Ks values for five tandem duplicate gene pairs (Table 1). We also analyzed segmental duplication (SD) events in the NAM genes. The duplication events of the NAM gene are illustrated in Figure 2. The results showed that 6.94% (10/144) of the NAM members exhibited segmental duplication, specifically on Chr 24 and Chr 35. The Ka/Ks ratio of all segmental duplication gene pairs was less than 1, implying that they were subject to negative selection. According to the duplication analysis of the NAM genes, some of these genes originated by tandem or segmental duplication. Such events may act as driving forces for gene evolution.

We further conducted a collinearity analysis of NAM genes in *O. hybridum* and NAM genes in *Arabidopsis thaliana* (L.) Heynh. (Figure 3), and the results are provided in Appendix A. The results indicate that the number of collinear gene pairs reached 3513, among which four pairs are NAM gene pairs: *AT4G28530.1*–*OhNAM105*, *AT5G13180.1*–*OhNAM073*, *AT5G13180.1*–*OhNAM135*, and *AT5G39610.1*–*OhNAM098*.

### 2.3. Conserved Motifs, Gene Structure, and Phylogenetic Analysis

To study the variety of the NAM gene structure, we extracted the exon–intron configuration information of the NAM genes from the annotation file. The exon–intron structure is depicted in Figure 4. The 144 NAM genes contained between 1 and 10 exons. Specifically, *OhNAM104* had only one exon, while *OhNAM113* had ten. The majority of the genes had 3 exons (77 genes, 53.47%), followed by 2 exons (43 genes, 29.86%). There were 9 genes containing 4 exons, 10 genes containing 5 exons, 6 genes containing 6 exons, and 2 genes containing 8 exons; genes with 7 exons and 9 exons each have one representation.

A total of 10 motifs were identified in the NAM family members. Figure 4 presents detailed information about these motifs. The results revealed that motif-1 to motif-6 were identified in almost all of the 144 NAM genes. However, motif-7 to motif-10 were present in only a few of the NAM genes; motif-7 was detected in 14 NAM genes, motif-8 was detected in 16 NAM genes, motif-9 was detected in 5 NAM genes, and motif-10 was detected in 7 NAM genes.

To explore the phylogenetic relationship among NAM proteins, the NAM protein sequences from *O. hybridum* and Arabidopsis were subject to multiple alignments using MEGA7.0, and the phylogenetic eVolution was determined. The 144 NAM genes were diVided into 14 subfamilies based on their gene structure (Figure 5). Notably, subfamily I, containing the most NAM genes, had 29 members. Subfamily VI, with the second largest number of NAM genes, had 28 members. Subfamilies XIII and XII, the smallest groups, each had 2 members. Interestingly, 14 NAM genes containing motif-7 were classified into subfamily-XIV, 5 NAM genes containing motif-9 were placed into subfamily-III, and 7 NAM genes containing motif-10 were classified into subfamily I. Thus, motifs within the same subfamily exhibit similar characteristics, while significant differences exist between subfamilies. Notably, all genes clustered in subfamily V and subfamily XII were NAM genes in *O. hybridum*, while the genes clustered in subfamily XIII belong to *A. thaliana*, which indicates that subfamily V and subfamily XII may be unique to *O. hybridum* and subfamily XIII may be unique to *A. thaliana*.

### 2.4. Differentially Abundant Metabolites in Pseudobulb at Different Stages

To elucidate the effects of endogenous hormones regulated by NAM genes on the growth of pseudobulbs, we first analyzed the differentially abundant metabolites (DAMs) of pseudobulbs at four different development stages. The results of the principal component analysis (PCA) indicated clear separation among the six comparisons across the four stages (Figure 6). Volcano plots were generated based on the DAMs data of the four stages, illustrating the differences between the various comparisons (Figure 7). The results indicate that the highest number of DAMs, totaling 431, was detected in the comparisons between the middle and late stages, with 344 metabolites being significantly up-regulated and 87 metabolites being significantly down-regulated. Notably, in this comparison, the number of significantly up-regulated metabolites is the highest among all comparisons, while the number of significantly down-regulated metabolites is the lowest among all comparisons. In the comparisons between the early and senescence stages, the fewest DAMs were detected, totaling 241, with 105 metabolites significantly up-regulated and 136 metabolites significantly down-regulated. In addition, in the comparison between the later and senescence stages, the fewest significantly up-regulated DAMs were detected, totaling 84. In the comparison between the early and middle stages, the highest number of significantly down-regulated DAMs was detected, totaling 291.

In order to clarify the metabolic pathways of DAMs, the KEGG database was utilized to annotate the DAMs; detailed information is provided in Appendix A. The KEGG analysis indicated that DAMs in the early vs. middle stages were significantly enhanced in stilbenoid, diarylheptanoid, and gingerol biosynthesis. DAMs in the early vs. later stages were significantly enhanced in the biosynthesis of unsaturated fatty acids. In the middle stage vs. senescence, the phenylpropanoid biosynthesis pathway was significantly enriched for DAMs. In the later vs. senescence stages, the biosynthesis of the unsaturated fatty acid pathway and the 2-Oxocarboxylic acid metabolism pathway were significantly enriched for DAMs. Notably, we found that jasmonic acid (JA) and abscisic acid (ABA) were enriched in the plant hormone signal transduction pathway in the comparisons of early vs. middle, middle vs. later, and later vs. senescence, and middle vs. later and middle vs. senescence, respectively. The results suggest that JA and ABA may be involved in the regulation of the growth and development of *O. hybridum* pseudobulbs. However, the matter of whether JA and ABA are regulated by members of the NAM gene family requires validation through correlation analysis.

### 2.5. Differentially Expressed Genes of NAM Gene Family

To determine how NAM genes regulate the growth of pseudobulbs, the expression levels of NAM genes at the four stages were analyzed using mRNA-seq. Based on the expression levels of NAM genes for each sample, we conducted a differentially expressed gene (DEG) analysis (Figure 8); the results indicated notable difference in the expression levels of certain NAM genes across the four different time periods. The largest number of DEGs of the NAM genes was identified in the contrast between middle and senescence, totaling 46, including 38 genes that were up-regulated and 8 genes that were down-regulated. The minimum number of DEGs of the NAM gene family were identified in later vs. senescence, only 19, including 16 genes that were up-regulated and 3 genes that were down-regulated. In middle vs. senescence, the highest number of up-regulated NAM genes were observed, totaling 38. In early vs. middle, there were the fewest up-regulated NAM genes—only 1. Meanwhile, during this period, the most down-regulated NAM genes were observed, totaling 28. In later vs. senescence, there were the least down-regulated genes, with a total of 3.

### 2.6. Expression Patterns of OhNAM Genes Based on RNA-Seq Data

The FPKM values calculated for the 144 NAM genes are provided in Appendix A. The log10 (FPKM + 1) values of the 144 OhNAM genes were systematically categorized and are displayed in a heat map (Figure 9). The results indicate significant variations in the expression patterns of the 144 OhNAM genes across the four stages. According to the expression profiles, we found that 14 OhNAM genes (*OhNAM089*, *126*, *025*, *119*, *142*, *042*, *073*, *113*, *012*, *136*, *055*, *079*, *111*, and *002*) consistently exhibited high expression levels across all four stages. The expression of eight NAM genes (*OhNAM007*, *009*, *047*, *083*, *098*, *049*, *030*, and *020*) is high only in the early stage. Six NAM genes (*OhNAM139*, *008*, *140*, *122*, *133*, and *010*) exhibit relatively high expression levels in both the early and middle stages. Five NAM genes (*OhNAM056*, *001*, *124*, *085*, and *115*) are exclusively highly expressed in the senescence stage. Notably, the transcripts of 17 OhNAM genes (*OhNAM003*, *031*, *033*, *066*, *067*, *068*, *069*, *080*, *082*, *084*, *086*, *091*, *116*, *132*, *134*, *141*, and *144*) were not detected in any of the four stages (FPKM value = 0).

### 2.7. Analysis of Differential Metabolites of Endogenous Hormones and the Corresponding Changes in Metabolically Related Genes

To investigate the molecular mechanism whereby OhNAM genes regulate the growth of *O. hybridum* pseudobulbs, we used PCC to analyze the correlation between differentially expressed OhNAM genes and two plant endogenous hormones identified through a metabolite analysis. The results of the analysis are provided in Appendix A, with summarized information presented in Table 2. The results indicate that JA and ABA are significantly correlated with several differentially expressed OhNAM genes during the growth and development of Oncidium pseudobulbs.

The relative content of JA at different stages and the expression levels of NAM genes associated with JA are both illustrated in Figure 10. The results indicate that the content of JA was significantly higher in the early stage than in the middle stage. Meanwhile, a total of 6 OhNAM genes (*OhNAM023*, *030*, *007*, *019*, *083*, and *047*) were significantly correlated with JA, and the expression levels of these genes were significantly down-regulated during this period. For middle vs. later, the content of JA in the middle stage was significantly lower than in the later stage. Meanwhile, the expression levels of two OhNAM genes (*OhNAM089* and *OhNAM025*) that are significantly correlated with JA were up-regulated. In later vs. senescence, the content of JA was significantly down-regulated, and the expression level of *OhNAM025*, which was significantly correlated with JA, also decreased significantly. We conducted the same analysis for other jasmonates (JAs) and correlated OhNAM genes (Figure 10). The results indicate that the content of all four JAs (Methyl jasmonate, Methyl Dihydrojasmonate, Prohydrojasmon, and (+)-Dihydrojasmonic acid) were significantly up-regulated in the middle vs. later analysis, and two OhNAM genes (*OhNAM089* and *OhNAM025*) were significantly correlated with them, with significantly up-regulated expression levels during this period.

The relative content of ABA and abscisic acid glucose ester (ABA-GE) at different stages and the expression levels of NAM genes associated with ABA and ABA-GE are both illustrated in Figure 10. In the analyses of the middle vs. later and middle vs. senescence stages, the content of ABA in the middle stage was notably lower than that in the senescence stage. Meanwhile, the expression levels of two (*OhNAM089*, *025*) NAM genes and three (*OhNAM119*, *055*, *136*) OhNAM genes significantly correlated with ABA were up-regulated. The variation in the relative content of ABA-GE is comparable to that of ABA, but with a continued increase in relative content from the late to senescence stages. Moreover, other than *OhNAM025*, all four OhNAM genes involved in regulating ABA are also implicated in the regulation of ABA-GE.

### 2.8. Verification of RNA-Seq Data by the qRT-PCR Assay

To validate the expression pattern of the NAM genes, we selected three crucial NAM genes for qPCR analysis. The qPCR results indicated that the relative expression trends of the three NAM genes were similar to those of FPKM values, suggesting that the RNA-seq data are reliable (Figure 11). The primer sequences designed using Primer Premier 5.0 software are provided in Appendix A.

## 3. Discussion

Analyzing the features of gene families enables researchers to gain a deeper understanding of the potential mechanisms by which plants regulate growth [15], and respond to various stresses [16], such as salt [17], cold [18], and drought [19], among other functions. Currently, the genome-wide identification of NAC TFs for various plants has been achieved [20]. However, there has been limited research on NAM proteins in the NAC gene family. Our research aimed to screen for NAM genes that may be involved in regulating pseudobulb growth in *O. hybridum* ‘Honeybee’; furthermore, we aimed to explain the roles and molecular processes of NAM genes in the growth of *O. hybridum*. In total, 144 NAM gene members with complete domains were found in *O. hybridum* following the bioinformatics analysis. When comparing the NAC genes identified in different kinds of plants, the number of NAM genes found in *O. hybridum* was similar to that in cotton (145) [21] and tobacco (154) [22], higher than that in pepper (104) [23] and grape (74) [24], but lower than that in alfalfa (421) [25] and wild wheat (200) [17]. The notable variations in the quantity of NAC and NAM genes among different plant species could be attributed to factors such as the genome size, gene duplication, and gene loss events. Our chromosomal localization analysis revealed that the 144 NAM genes are unevenly distributed across the 39 chromosomes during biological evolution. Based on the characteristics of these NAM genes, the 144 NAM genes were classified into 14 subfamilies. We observed that NAM genes showing similar motif arrangements and gene structures were predominantly classified into the same subfamily. For instance, motif-7 was identified exclusively in subfamily XIV and motif-9 was identified only in subfamily III. This suggests that the significant structural conservation among the NAM genes in *O. hybridum* and the NAM genes shared similar functional characteristics during the process of biological evolution. The results also prove the reliability of the phylogenetic tree constructed in this study.

Gene duplication and the retention of duplicate gene pairs at a high rate have led to a substantial number of duplicate genes in plant genomes. The emergence of new functions has been aided by these gene duplication events, such as the development of floral structures [26], the enhancement of disease resistance [27], and adaptation to stress [28]. In this research, five pairs of segment duplication genes and sixteen pairs of tandem duplication genes were discovered; these duplication events may have led to the expansion of the NAM genes in *O. hybridum*. Paired NAM genes exhibit structural similarity but may differ in terms of functionality. Therefore, further validation is required. Furthermore, we also calculated the Ka/K values for these duplicated NAM genes pairs. In genetics, the Ka/Ks value can indicate whether there is selective pressure acting on these protein-coding genes [29]. We discovered that the Ka/Ks values of the duplicated NAM gene pairs examined in this study are all less than 1, which indicates that all NAM genes in *O. hybridum* have undergone negative selection. Additionally, by conducting a collinearity analysis between *O. hybridum* and *A. thaliana*, we identified four pairs of NAM genes, indicating that these gene pairs are conserved between the two species. This conservation suggests that NAM genes may play an important role in the biological functions of both species.

Interestingly, we found that the expression of NAM gene family members in *O. hybridum* exhibits stage-specific patterns, suggesting potential functional differences in NAM genes across the four stages of *O. hybridum* pseudobulb growth and development. Tan et al. discovered that NAC genes may be involved in the control of rose seedlings, leading to stage-specific changes in different seedling ages [30]. Gu et al. discovered the differential expression level of *LiNAC8* and *LiNAC13* at different growth stages in *Lagerstroemia indica* L., and these two NAC genes are involved in the control of weeping traits [31]. It is notable that we found 17 members of the NAM gene family (*OhNAM003*, *031*, *033*, *066*, *067*, *068*, *069*, *080*, *082, 084*, *086*, *091*, *116*, *132*, *134*, *141*, and *144*) that are not expressed at any stage. The lack of expression may be attributed to the tissue-specific expression patterns of these 17 NAM genes in different tissues of *O. hybridum*. Additionally, we observed significant differences in the accumulation of certain metabolites at different developmental stages in *O. hybridum*. The results of the KEGG analysis show that these DAMs are significantly enriched in four metabolic pathways. These pathways have also been observed in similar studies. For instance, Liu et al. conducted a KEGG analysis of DAMs at different developmental stages of *Zingiber officinale* Roscoe rhizomes, successfully identifying the stilbenoid, diarylheptanoid, and gingerol biosynthesis pathways [32]. The metabolites enriched in these four metabolic pathways may be related to plant growth and development. For example, oleic acid, linoleic acid, and erucic acid are enriched in the biosynthesis of the unsaturated fatty acid pathway. Meï C. et al. found an association between the growth rate of cell populations and the degree of unsaturation of fatty acids with 18 carbons. The levels of polyunsaturated fatty acids may partially depend on the rate of cell division [33]. To gain deeper insights into the relationship between NAM DEGs and endogenous hormones, as well as the role of the NAM gene family in the growth of *O. hybridum*, we performed a combined analysis using both metabolomic and transcriptomic data. Among all the DAMs that were significantly associated with the NAM genes in *O. hybridum*, we found two types of endogenous plant hormones: JA and ABA.

Earlier research has demonstrated that moderate concentrations of JA have a growth-promoting effect on plants. Wei et al. found that appropriate concentrations of JA can enhance the germination rate and sprout length of alfalfa seeds [34]. Satora K et al. found that the concentration of JA increases as grape seeds mature. When seeds from 18 to 28 days after flowering were used as explants and treated with 0.45 mol/L of JA, it induced callus formation, indicating that exogenous JA can stimulate cell division in grapes [35]. In this research, we found that the content of JAs is relatively high in the early stages of *O. hybridum*. This period corresponds to the primary growth phase of the orchid. The early presence of JAs may promote the growth and development of *O. hybridum*. Furthermore, research indicates the NAC gene family has a regulatory function in JA signaling; Bu et al. discovered that the *A. thaliana* NAC family members *ANAC019* and *ANAC055* could act as transcription activators to control the JA-induced expression of defense genes [36]. In this study, we detected six NAM genes that were significantly associated with JA, including *OhNAM023*, *030*, *007*, *019*, *083*, and *047*. Furthermore, we identified two NAM genes that were significantly associated with other four JAs: *OhNAM089* and *OhNAM025*. These NAM genes may regulate the growth of *O. hybridum* by modulating the metabolism of JAs.

It has been found that ABA promotes plant senescence; impressive progress has been made in clarifying the ABA metabolism and signaling pathways during plant growth and development. Previous research indicates that NAC genes have a regulatory function in ABA signaling. Mao et al. found that the ectopic expression of *OsNAC2* upregulates the expression of the ABA biosynthesis genes *OsNCED3* and *OsZEP1* while downregulating the expression of the ABA catabolism gene *OsABA8ox1*, resulting in an elevation of the ABA content. The overexpression of *OsNAC2* results in the upregulation of the chlorophyll degradation genes *OsSGR* and *OsNYC3*, thereby accelerating the senescence of leaves [37]. In this study, we found that the ABA content in *O. hybridum* was relatively high during the late and senescence stages, indicating that the orchids are undergoing senescence during these periods. We detected five NAM genes that were significantly associated with ABA, namely, *OhNAM089*, *025*, *119*, *055*, and *136*. These genes may promote the senescence of *O. hybridum* by regulating the metabolism of ABA. Furthermore, earlier research has shown that MeJA can promote the synthesis of ABA. Wang et al. found that MeJA can stimulate the synthesis of ABA in *Cucurbita pepo* L. under cold conditions [38]. Creelman et al. found that the content of ABA and JA in soybean leaves significantly increases after water stress treatment, with JA accumulating one to two hours earlier than ABA [39]. In this research, we observed that both ABA and MeJA reach high levels during the late stage. Thus, we speculate that the increase in MeJA content may be one of the factors for the induction of ABA content in *O. hybridum*. It is worth noting that we observed both *OhNAM089* and *OhNAM025* to be significantly correlated with JAs as well as ABA. The conclusion that a single NAC gene is associated with multiple plant hormones has also been reached in studies of other NAC gene families. Pei et al. discovered that ethylene controls cell proliferation by finely adjusting the miRNA164/RhNAC100 module. Furthermore, their study revealed that the equilibrium between *RhNAC100* and miR164 could also be affected by various plant hormones such as auxin and gibberellin [40]. Interestingly, the relative content of ABA and ABA-GE shows different trends in the later vs. senescence stages. During this period, the relative content of ABA tends to stabilize, whereas the relative content of ABA-GE significantly increases. This may be due to *OhNAM025* positively regulating ABA, whereas the regulation of ABA-GE is not mediated by this gene.

Overall, eleven candidate NAM genes are significantly correlated with two endogenous hormones, suggesting that the NAM gene family is crucial for controlling plant growth at different stages. These eleven NAM genes could yield important insights into *O. hybridum* development and genetic research. Based on the findings of this study, future research could explore the functions and mechanisms of these genes in greater depth. For instance, we determined that *OhNAM089* and *OhNAM025* are involved in the regulation of JA and ABA, which prompts us to consider the possibility of multigene cooperative regulation by these transcription factors [41]. Consequently, further investigation into their collective roles in pseudobulb development would be warranted. Moreover, research indicates a correlation between pseudobulb formation and flowering in Orchidaceae plants [2]. Therefore, the phenotypic analysis of transgenic plants could be utilized to investigate the regulatory mechanisms of these NAM genes in different tissues of *O. hybridum*. This study also provides candidate target genes for the genetic improvement of Orchidaceae plants. Future research could involve identifying homologous genes in other Orchidaceae species to investigate whether these NAM genes have similar functions in JA and ABA regulation. Additionally, gene editing or transgenic approaches could be employed to enhance the resistance or adaptability of Orchidaceae plants [42,43,44].

## 4. Materials and Methods

### 4.1. Identification of NAM Genes in O. hybridum

The genome sequence of *Oncidium hybridum* ‘Honeybee’ was assembled by our research group and uploaded to the Genome Warehouse in the National Genomics Data Center under the accession number GWHDRIS00000000 (https://ngdc.cncb.ac.cn/gwh/Assembly/66233/show, accessed on 10 August 2023). The gene annotation information was annotated but not officially published. To identify the NAM genes in *O. hybridum*, we employed two approaches. First, we downloaded the Hidden Markov Model (HMM) of the NAM domain (PF02365) from the Pfam database and searched for NAM genes in *O. hybridum* using an HMM search (E < 1 × 10^−5^) [45]. Then, to validate the genes obtained through the HMM search, the BLASTP method was employed to identify the NAM gene family using TBtools (version 2.119) [46]. Subsequently, we took the intersection of the NAM genes obtained using the two methods and uploaded all NAM protein sequences to NCBI Batch CD-Search (https://www.ncbi.nlm.nih.gov/Structure/bwrpsb/bwrpsb.cgi, accessed on 17 March 2024) to ensure that all sequences contain the NAM domains. Furthermore, the sequences of the NAM proteins were uploaded to the ExPASy platform (https://web.expasy.org/protparam/ accessed on 17 March 2024) to calculate their molecular weights, isoelectric points (pI), and amino acid numbers [47]. The websites WoLF PSORT (https://wolfpsort.hgc.jp/, accessed on 17 March 2024) and Plant-mPLoc (http://www.csbio.sjtu.edu.cn/bioinf/plant-multi/, accessed on 30 August 2024) were utilized to calculate the subcellular localization [48,49].

### 4.2. Chromosomal Localization, Gene Duplication, and Collinearity with A. thaliana

The locations of the 144 NAM genes were obtained from the gff file. All NAM genes were mapped to the chromosomes of *O. hybridum* using the TBtools Gene Location Visualization tool (version 2.119). The collinearity analysis within the *O. hybridum* genome and between the *O. hybridum* and *A. thaliana* was conducted using the Multiple Collinearity Scan Toolkit X (MCScanX, version 1) [50]. The 144 NAM protein sequences were compared with each other using BLASTP, and the comparison results, along with a file containing the gene positions of *O. hybridum*, were input into MCScanX to determine the duplication types, utilizing the default parameters [51]. TBtools Advanced Circos (version 2.119) was used to depict the collinearity correlations within the *O. hybridum* genome. For every duplicated NAM gene pair, the KaKs_Calculator software (version 1) was utilized to calculate the values of non-synonymous (Ka) substitution rates and synonymous (Ks) substitution rates [52]. The genome and annotation data of *A. thaliana* used for the collinearity analysis were downloaded from the TAIR database (http://www.arabidopsis.org, accessed on 31 August 2024). A TBtools Dual Synteny Plot (version 2.119) was used to depict the collinearity correlations between the *O. hybridum* and *A. haliana* genomes.

### 4.3. Conserved Motifs, Gene Structure, and Phylogenetic Trees Analysis

The conserved motifs of the NAM proteins were predicted utilizing the MEME platform (http://meme-suite.org/, accessed on 17 March 2024). The parameters were configured to identify up to 10 motifs for each NAM gene, with the optimal width range set between 6 and 50 [53]. The exon and intron information was extracted from the gff file, and the structure of each NAM gene was visualized using TBtools. In order to investigate the phylogenetic relationships of NAM proteins between *O. hybridum* and the ortholog in *A. thaliana*, a total of 37 Arabidopsis NAM amino acid sequences were downloaded from the TAIR database (http://www.arabidopsis.org, accessed on 17 March 2024) to construct the NAM phylogenetic trees [54]. The sequence alignment of the OhNAM proteins and Arabidopsis NAM proteins was conducted using the MUSCLE program, and the parameters were set to the default values [55]. The alignment results were utilized to generate a phylogenetic tree using the neighbor-joining (NJ) approach in MEGA software (version 7), employing the JTT model [56].

### 4.4. Plant Materials and Growth Condition

The *O. hybridum* ‘Honeybee’ samples utilized in this study were obtained from Hainan Boda Orchid Technology Co., Ltd. (110°14′33.1′′ E,19°47′29.7′′ N), Haikou, China. All *O. hybridum* samples were planted in a greenhouse with summer temperatures ranging from 25 to 42 °C, winter temperatures from 20 to 30 °C, annual relative humidity between 70% and 80%, and an illumination intensity between 20,000 and 30,000 lux. We selected three specimens of the plant from each of the early, middle, later, and senescence stages, and the pseudobulb tissue was systematically collected from each sample. The experimental samples are shown in Figure 12.

### 4.5. Expression Patterns and Differentially Expressed Gene Analysis

The total RNA of *O. hybridum* was extracted using the RNA extraction kit (Qiagen, Hilden, Germany), with three biological replicates performed for each stage of pseudobulb development. Next, the quality of the RNA was evaluated using the Agilent Bioanalyzer 2100 system (Agilent Technologies, Santa Clara, CA, USA). Subsequently, the reverse transcription kits were utilized to reverse transcribe the RNA into cDNA. Then, the Illumina Novaseq platform (Illumina, San Diego, CA, USA) was used to sequence the generated cDNA libraries. To obtain high-quality clean data, the raw RNA sequence data were filtered using Fastp by removing reads containing adapters or poly-N regions and low-quality reads [57]. The gene model annotation files and reference genome of *O. hybridum* ‘Honeybee’ are detailed in Section 4.1 above. The HISAT2 was utilized to align the clean data with the reference genome of *O. hybridum* ‘Honeybee’ graph-based genome alignment and genotyping with HISAT2 and the HISAT-genotype [58]. FeatureCounts (version 2.0.1) was employed to count reads for each gene [59]. Then, the fragments per kilobase of transcript per million mapped reads (FPKM) were calculated based on the gene length and read count mapped to each gene; this is currently the most widely applied technique for estimating gene expression levels. The raw RNA sequence data of pseudobulbs in *O. hybridum* were uploaded to the Genome Sequence Archive [60] in the National Genomics Data Center [61], China National Center for Bioinformation/Beijing Institute of Genomics, Chinese Academy of Sciences (https://bigd.big.ac.cn/gsa/browse/CRA017038, accessed on 14 June 2024).

In order to identify the NAM genes with significantly different expression levels across the four stages, we conducted a statistical analysis of the expression data, employing the DESeq2 R package, which is based on a negative binomial distribution model [62]. The criterion we employed, (log2(FoldChange)| ≥ 1 & padj ≤ 0.05), is commonly used for differentially expressed gene (DEG) selection, and genes meeting this criterion are considered DEGs. Subsequently, from the resulting list of DEGs, we further screened and identified 144 OhNAM genes relevant to our experimental study. Finally, a heatmap analysis was used to visualize the expression patterns of the 144 OhNAM genes over the four distinct stages. Due to the large differences in FPKM values among the 144 OhNAM genes, log10 (FPKM + 1) values were utilized to generate the heatmap; this allows for a better visualization of the differences in gene expression levels and makes it easier to observe genes with lower expression levels on the heatmap [63].

### 4.6. Differential Accumulated Metabolite and Endogenous Plant Hormones Analysis

The metabolites of each sample were determined by Novogene Bioinformatics Technology Co., Ltd. (Beijing, China). In brief, 100 mg samples of pseudobulb tissue were extracted with 80% methanol. Three biological replicates were set for each sample. Then, the samples were subjected to centrifugation at 15,000× *g* and 4 °C for 20 min. Subsequently, the supernatant was diluted with LC-MS-grade water to achieve a final concentration of 53% methanol, and then it was centrifuged again under the same parameters. Finally, the resulting supernatant was analyzed using an LC-MS/MS system under the following conditions: HPLC column, Hypersil Gold column (100 × 2.1 mm, 1.9 μm); solvent system, positive polarity mode: 0.1% FA in water (A), methanol (B), negative polarity mode: 5 mMammonium acetate, pH 9.0 (A), methanol (B); solvent gradient, 2% B, 1.5 min, 2–85% B, 3 min, 85–100% B, 10 min, 100–2% B, 10.1 min, 2% B, 12 min; spray voltage, 3.5 kV; sheath gas flow rate, 35 psi; aux gas flow rate, 10 L/min; capillary temp, 320 °C; S-lensRFlevel, 60; aux gas heater temp, 350 °C. The raw data were subjected to qualitative and quantitative analysis using Compound Discoverer 3.1.

Based on the accurate metabolite data, principal component analysis (PCA) and partial least squares discriminant analysis (PLS-DA) were conducted to calculate the variable importance in projection (VIP) value, employing metaX [64]. Subsequently, we calculated the fold change (FC) and *p*-value. Then, metabolites with a VIP value greater than 1.0 and a fold change exceeding 1.5, or a fold change less than 0.667, along with a *p*-value below 0.05, were categorized as differentially accumulated metabolites (DAMs). The functions and pathways of the DAMs were investigated using the KEGG database, and enrichment analysis was carried out on these DAMs. If x/N was greater than y/N and the *p*-value was less than 0.05, the metabolic pathway was deemed to be significantly enriched [65].

### 4.7. Transcriptome and Metabolome Correlation Analysis of the NAM Gene Family

In order to elucidate the regulatory mechanisms of the NAM gene family in the growth and development of pseudobulbs in *O. hybridum*, a correlation analysis between DEGs and DAMs was conducted. First, the degree of correlation between DEGs and DAMs was assessed using the Pearson correlation coefficient (PCC); DEGs and DAMs with PCC > 0.80 and *p*-value < 0.05 were considered to have a significant correlation [66]. A negative correlation was assigned when the coefficient was less than 0, whereas a positive correlation was designated when it exceeded 0. Subsequently, specific endogenous plant hormones were selected from the DAMs, and differentially expressed NAM genes significantly correlated with these hormones were identified.

### 4.8. Validation of Quantitative Real-Time PCR

To determine the accuracy of RNA-seq results, qRT-PCR was performed on three NAM genes related to the development of pseudobulbs. Total RNA was extracted from pseudobulbs at four stages using the Total RNA Extractor (Trizol) (Sangon, Shanghai, China). Three replicates were set for each sample. Maxima Reverse Transcriptase (Thermo Scientific, Waltham, MA, USA) was utilized to reverse transcribe RNA into cDNA. Primer Premier 5.0 was used to design specific primers for qRT-PCR analysis. The qRT-PCR was conducted in a 20 µL reaction mixture containing 2 µL cDNA, 7.2 µL double-distilled H_2_O, 0.8 µL primer mix, and 10 µL SybrGreen qPCR Master Mix (Sangon, Shanghai, China) on an ABI7500 system. Relative gene expression levels were calculated using the 2−ΔΔCT method with β-actin as the internal reference gene.

## 5. Conclusions

In total, 144 NAM proteins in *O. hybridum* were successfully discovered in this investigation. We conducted comprehensive investigations of their physicochemical properties, chromosomal locations, gene duplication, gene structure, conserved motifs, and phylogenetic relationships. Moreover, we meticulously explored the differential expression patterns of NAM genes and DAMs across the four developmental stages of pseudobulbs, completing both transcriptomic and metabolomic analyses. By conducting a correlation analysis between DAMs and DEGs, we identified eight NAM genes associated with the JA metabolism and five NAM genes associated with the ABA metabolism. These results not only bridge the knowledge gap regarding the NAM gene family of *O. hybridum*, but may also have a significant impact on the genetic improvement and elucidation of the molecular mechanism underlying pseudobulb growth and development.

## Figures and Tables

**Figure 1 ijms-25-10355-f001:**
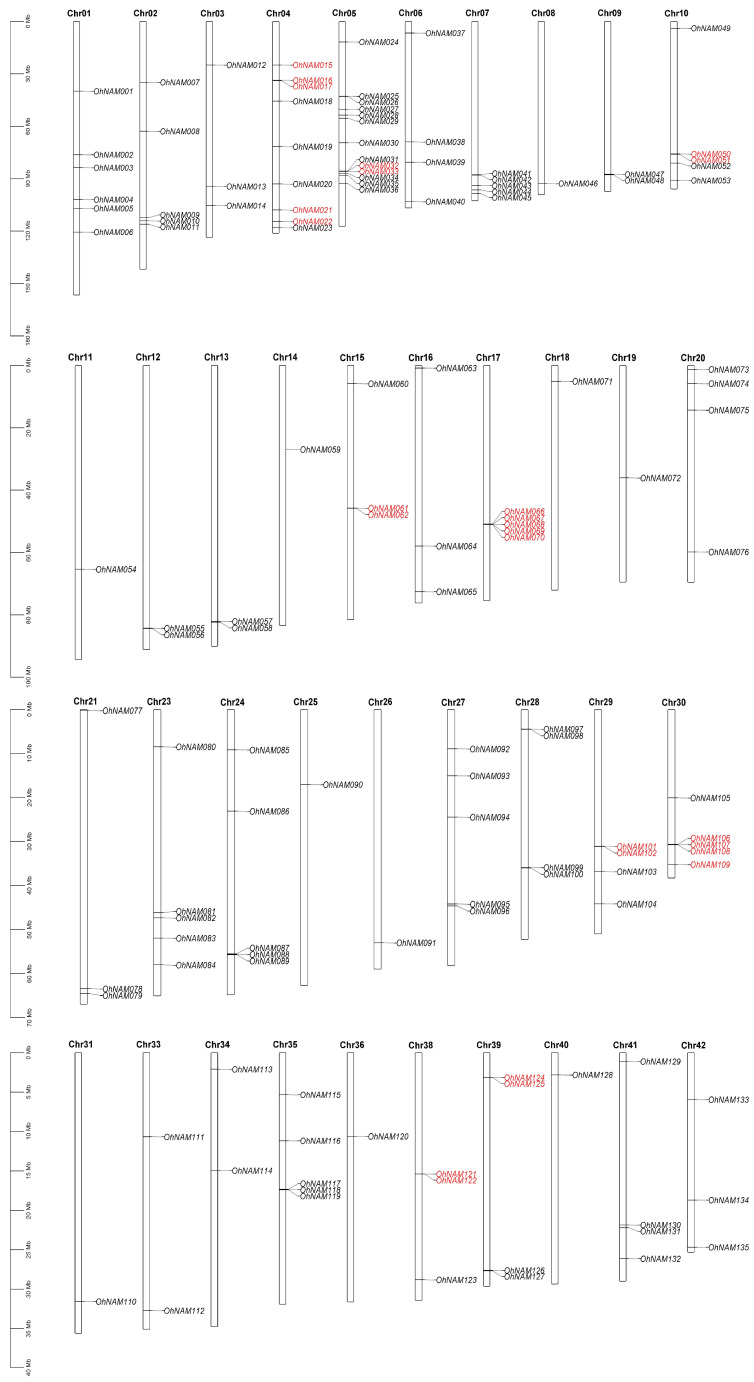
Chromosomal locations of the 144 NAM genes. The red font indicates pairs of tandem duplicated genes.

**Figure 2 ijms-25-10355-f002:**
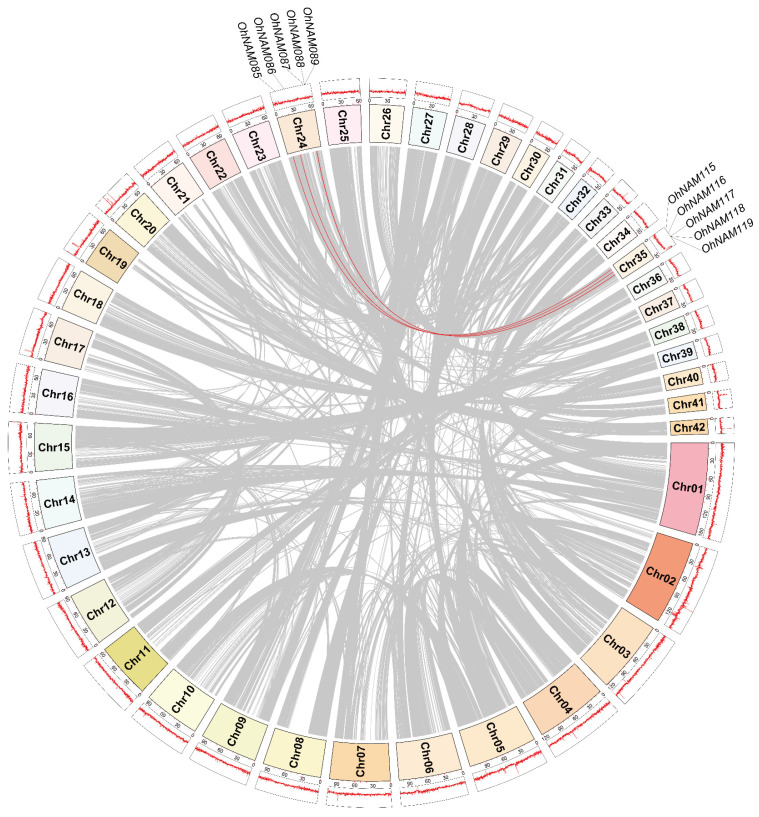
Segmental duplication of NAMs. From the inner to outer parts, the tracks represent the collinearity relationships of all whole-genome genes, the chromosome names, the GC ratio, and the repeated NAM gene location. Gray lines indicate collinear blocks in *O. hybridum*, while repeated NAM gene pair segments are connected by red lines. Each chromosome is represented by a color-coded block based on its length. When the line representing the GC ratio moves further from the center, it indicates a higher ratio. Segmental duplicated NAM gene pairs are annotated on the corresponding GC ratio diagram based on their positions on the chromosomes.

**Figure 3 ijms-25-10355-f003:**
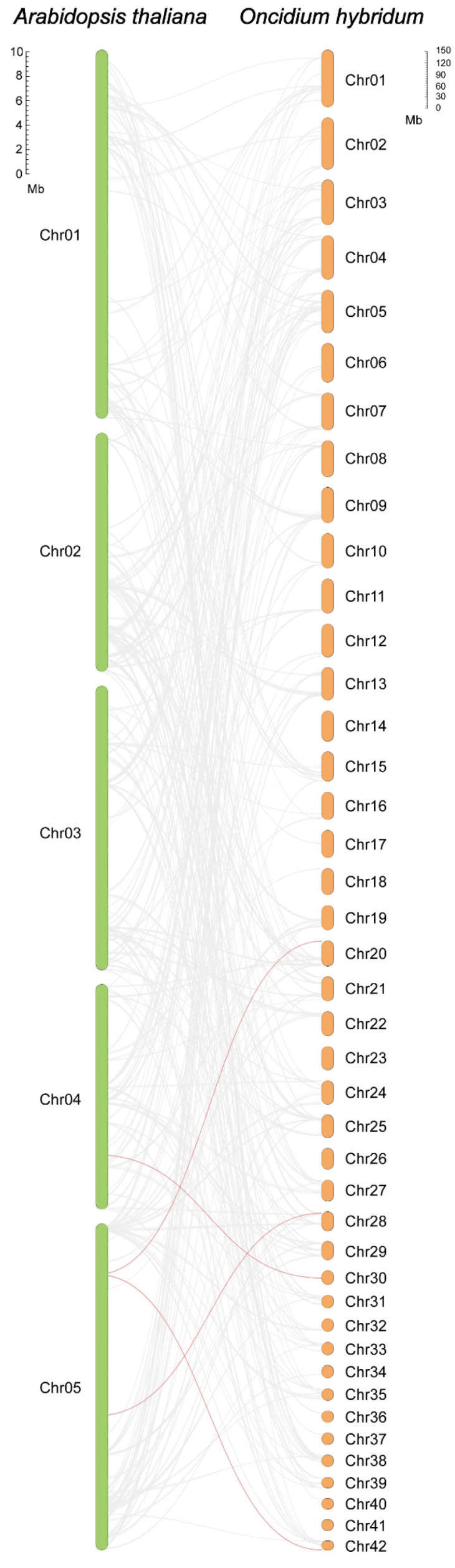
Collinearity analysis between *O. hybridum* and *A. thaliana*. The chromosomes of *A. thaliana* are represented by green blocks, while the chromosomes of *O. hybridum* are represented by orange blocks. Gray lines represent the collinear blocks between the two species, while the collinear NAM gene pairs are connected by red lines.

**Figure 4 ijms-25-10355-f004:**
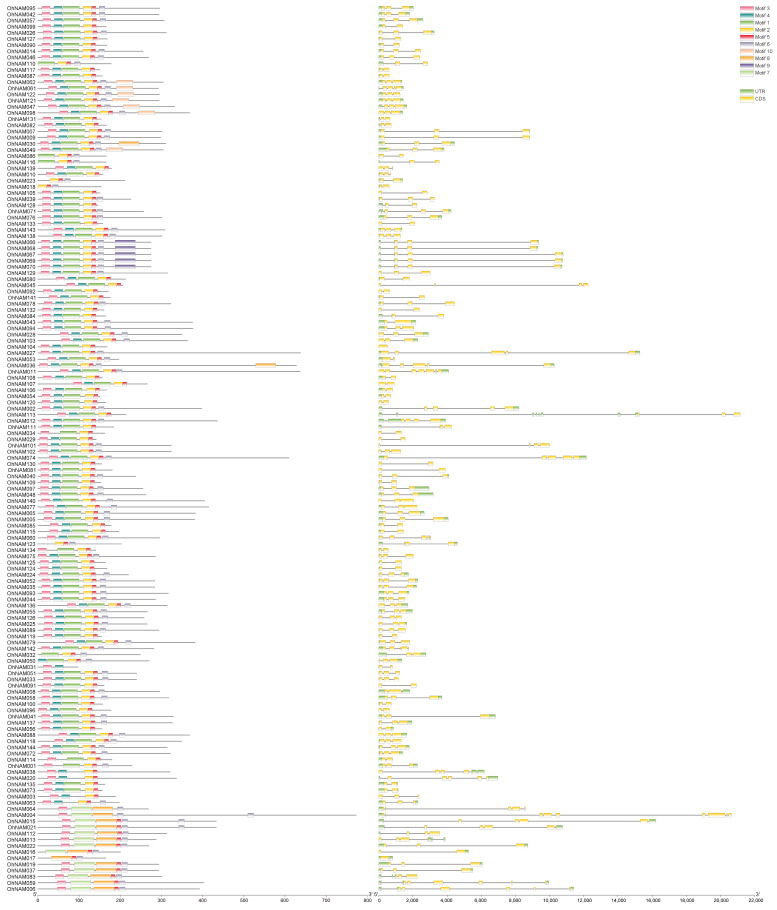
Conserved motif and gene structure analysis of NAM genes in *O. hybridum*. Ten distinct colors are used to indicate the ten identified motifs. The UTR is marked with green and the CDS is marked with yellow.

**Figure 5 ijms-25-10355-f005:**
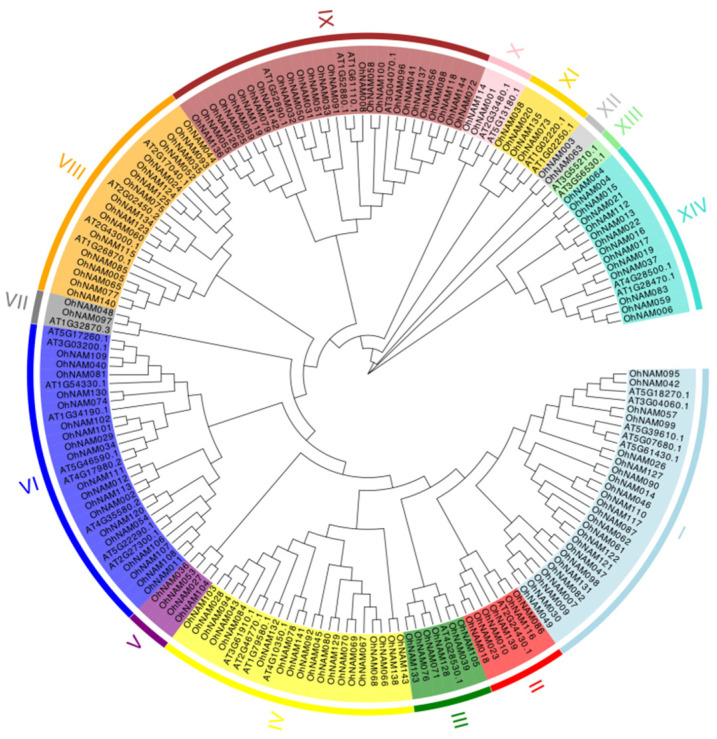
Phylogenetic tree of NAM gene family members in *O. hybridum* and *A. thaliana*. The distance between branches represents the phylogenetic relationship between NAM genes. All NAM genes were classified based on a motif structure analysis. Members divided into the same subfamily are labeled with the same color.

**Figure 6 ijms-25-10355-f006:**
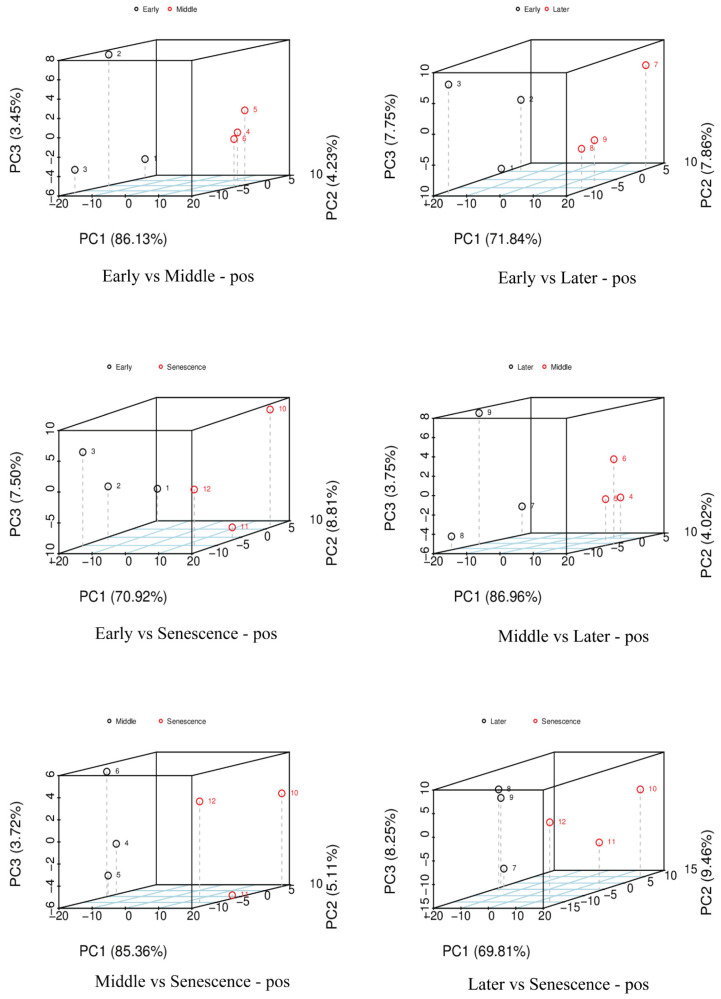
Results of the PCA of DAMs in a positive ion mode; the PCA in a negative ion mode is provided in Appendix A. Samples 1–3 represent the early stage, 4–6 represent the middle stage, 7–9 represent the late stage, and 10–12 represent the senescence stage.

**Figure 7 ijms-25-10355-f007:**
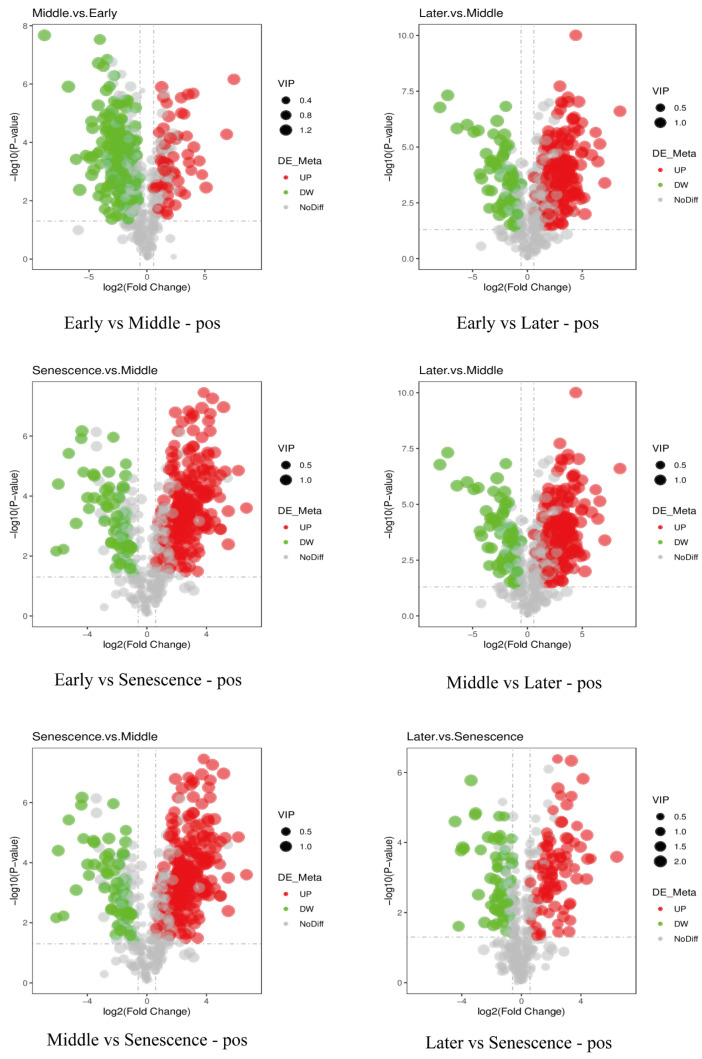
Volcano plots of DAMs across four stages in the positive ion mode. Volcano plots of DAMs in the negative ion mode are provided in Appendix A. Significantly down-regulated metabolites are shown by the green point, while the red point represents significantly up-regulated metabolites. The VIP value is indicated by the size of the point.

**Figure 8 ijms-25-10355-f008:**
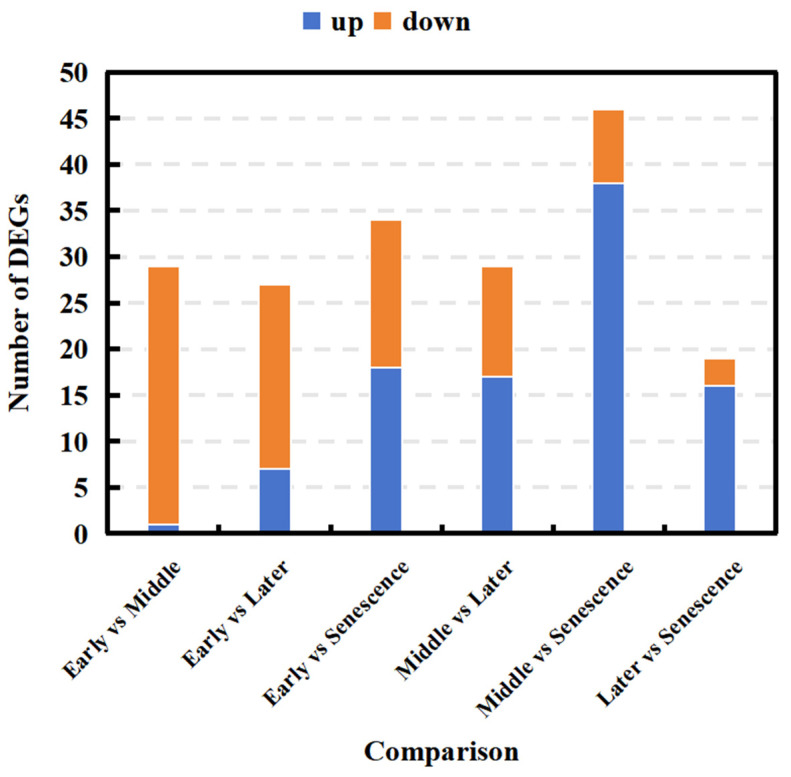
DEGs of NAM gene family members in *O. hybridum* at different stages. Blue indicates up-regulated DEGs, while orange indicates down-regulated DEGs.

**Figure 9 ijms-25-10355-f009:**
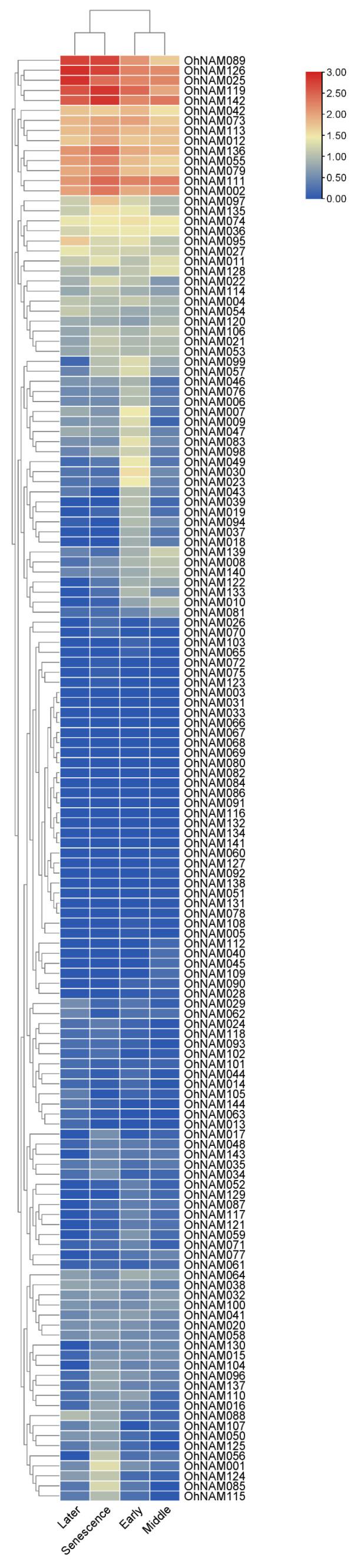
Expression heatmap of NAM genes. The numbers on the color scale represent the log10 (FPKM + 1) values, and, based on these values, the relative expression levels of each gene are marked from red (high) to blue (low).

**Figure 10 ijms-25-10355-f010:**
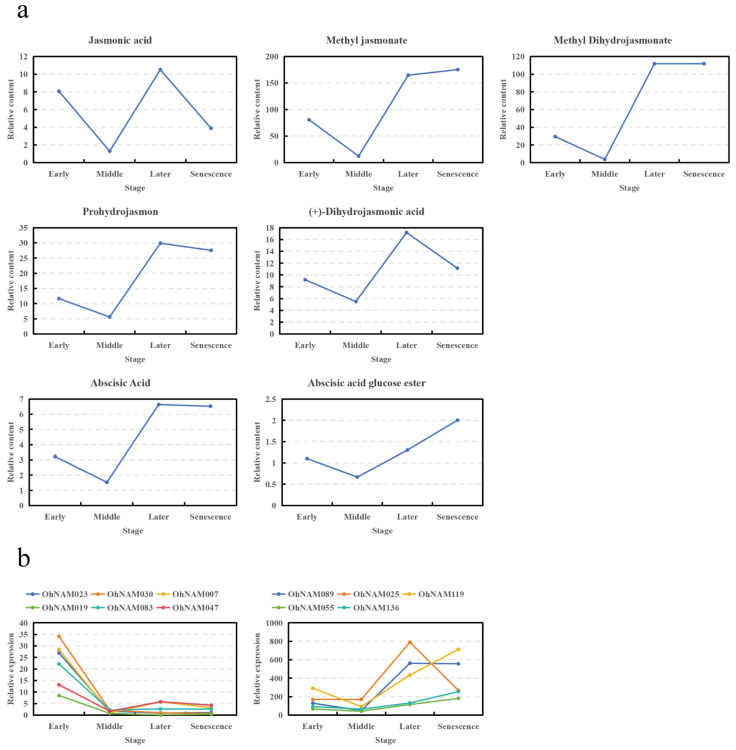
Relative content of JAs, ABA, and ABA-GE and expression levels (FPKM) of NAM genes associated with JAs, ABA, and ABA-GE during four stages: (**a**) the relative contents of JAs, ABA, and ABA-GE; (**b**) the expression levels (FPKM) of NAM genes significantly associated with JAs, ABA, and ABA-GE are represented in two separate line graphs, with different colors used to indicate different genes in each graph.

**Figure 11 ijms-25-10355-f011:**
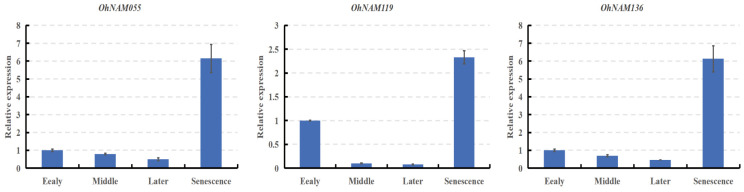
Quantitative real-time PCR analysis of the three NAM genes across the four development stages. The blue bars represent the relative expression, and the error bars indicate the positive and negative standard deviations.

**Figure 12 ijms-25-10355-f012:**
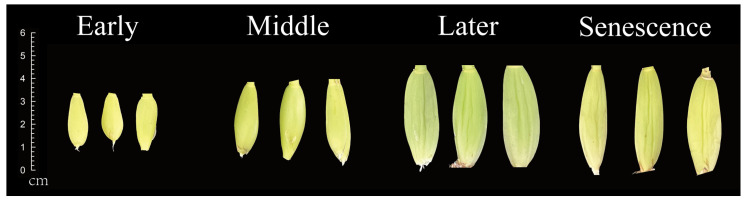
*O. hybridum* pseudobulbs in the early, middle, later, and senescence stages.

**Table 1 ijms-25-10355-t001:** Ka, Ks, and Ka/Ks of duplicated pairs of the NAM gene family.

Duplicated Gene Pairs	Nonsynonymous (Ka)	Synonymous (Ks)	Ka/Ks	Type
*OhNAM085* and *OhNAM115*	0.04	0.19	0.20	segmental
*OhNAM086* and *OhNAM116*	0.02	0.11	0.23	segmental
*OhNAM087* and *OhNAM117*	0.02	0.21	0.11	segmental
*OhNAM088* and *OhNAM118*	0.07	0.22	0.30	segmental
*OhNAM089* and *OhNAM119*	0.02	0.22	0.12	segmental
*OhNAM015* and *OhNAM016*	0.47	-	-	tandem
*OhNAM016* and *OhNAM017*	0.06	0.15	0.39	tandem
*OhNAM021* and *OhNAM022*	0.48	-	-	tandem
*OhNAM032* and *OhNAM033*	0.38	-	-	tandem
*OhNAM050* and *OhNAM051*	0.31	-	-	tandem
*OhNAM061* and *OhNAM062*	0.05	0.16	0.34	tandem
*OhNAM066* and *OhNAM067*	0.00	0.03	0.30	tandem
*OhNAM067* and *OhNAM068*	0.00	0.02	0.37	tandem
*OhNAM068* and *OhNAM069*	0.01	0.02	0.44	tandem
*OhNAM069* and *OhNAM070*	0.00	0.00	-	tandem
*OhNAM101* and *OhNAM102*	0.00	0.00	0.91	tandem
*OhNAM106* and *OhNAM107*	0.12	0.15	0.81	tandem
*OhNAM107* and *OhNAM108*	0.13	0.14	0.89	tandem
*OhNAM108* and *OhNAM109*	0.37	2.36	0.15	tandem
*OhNAM121* and *OhNAM122*	0.04	0.22	0.21	tandem
*OhNAM124* and *OhNAM125*	0.03	0.09	0.37	tandem

**Table 2 ijms-25-10355-t002:** PCC and *p*-value of correlation pairs of NAM differentially expressed genes with differentially abundant endogenous hormones.

Comparison	Metabolite	Gene	PCC	*p*-Value
Early vs. middle	Jasmonic acid	*OhNAM007*	0.85	0.03
Early vs. middle	Jasmonic acid	*OhNAM019*	0.99	0.00
Early vs. middle	Jasmonic acid	*OhNAM023*	0.99	0.00
Early vs. middle	Jasmonic acid	*OhNAM030*	0.83	0.04
Early vs. middle	Jasmonic acid	*OhNAM047*	0.91	0.01
Early vs. middle	Jasmonic acid	*OhNAM083*	0.83	0.02
Middle vs. later	Jasmonic acid	*OhNAM025*	1.00	0.00
Middle vs. later	Jasmonic acid	*OhNAM089*	0.99	0.00
Middle vs. later	Methyl jasmonate	*OhNAM025*	1.00	0.00
Middle vs. later	Methyl jasmonate	*OhNAM089*	0.99	0.00
Middle vs. later	Methyl dihydrojasmonate	*OhNAM025*	0.99	0.00
Middle vs. later	Methyl dihydrojasmonate	*OhNAM089*	1.00	0.00
Middle vs. later	Prohydrojasmon	*OhNAM025*	0.94	0.00
Middle vs. later	Prohydrojasmon	*OhNAM089*	0.98	0.00
Middle vs. later	(+)-Dihydrojasmonic acid	*OhNAM025*	0.95	0.00
Middle vs. later	(+)-Dihydrojasmonic acid	*OhNAM089*	0.92	0.01
Later vs. senescence	Jasmonic acid	*OhNAM025*	0.98	0.00
Middle vs. later	Abscisic acid	*OhNAM025*	1.00	0.00
Middle vs. later	Abscisic acid	*OhNAM089*	0.98	0.00
Middle vs. senescence	Abscisic acid	*OhNAM055*	0.98	0.00
Middle vs. senescence	Abscisic acid	*OhNAM119*	0.98	0.00
Middle vs. senescence	Abscisic acid	*OhNAM136*	0.98	0.00
Middle vs. senescence	Abscisic acid glucose ester	*OhNAM089*	0.93	0.01
Middle vs. senescence	Abscisic acid glucose ester	*OhNAM055*	0.97	0.00
Middle vs. senescence	Abscisic acid glucose ester	*OhNAM119*	0.90	0.01
Middle vs. senescence	Abscisic acid glucose ester	*OhNAM136*	0.97	0.00

## Data Availability

The data generated or analyzed in this study are included in this article and its additional materials. Genomic data are deposited in the Genome Warehouse in National Genomics Data Center under the accession number GWHDRIS00000000 (https://ngdc.cncb.ac.cn/gwh/Assembly/66233/show, accessed on 10 August 2023). The raw RNA sequence data are uploaded to the Genome Sequence Archive in the National Genomics Data Center under the accession number PRJCA018867 (https://bigd.big.ac.cn/gsa/browse/CRA017038, accessed on 14 June 2024).

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
