# Peer review of "Integrative Analysis of Transcriptome and Metabolome Reveals the Pivotal Role of the NAM Family Genes in Oncidium hybridum Lodd. Pseudobulb Growth"

_ijms, 2024, doi:10.3390/ijms251910355_

Round 1
Reviewer 1 Report
Comments and Suggestions for Authors
Introduction:
The introduction provides a good overview of the importance of Oncidium hybridum and the role of the NAM gene family. However, it could benefit from a more detailed review of existing literature, particularly studies that have investigated the regulation of pseudobulb growth in other orchid species. This would help contextualize the study's contributions more clearly.
Methods:
While the methods are generally well described, there are areas where additional detail would enhance reproducibility. For example, more information on the statistical methods used for data analysis would be beneficial. Additionally, it would be helpful to clarify any assumptions made during the bioinformatic analyses.
Results:
The results are clearly presented, but some of the figures could be enhanced with more detailed legends to help guide the reader through complex data. For example, in the figures related to gene duplication and phylogenetic trees, additional annotations could improve comprehension.
Conclusion:
The conclusions are well supported by the results. However, the discussion could be expanded to include potential future research directions based on the findings, as well as the broader implications for the field of orchid genetics.
References:
The references cited are appropriate and relevant to the research. However, the addition of more recent studies, particularly those published in the last two years, could strengthen the manuscript.
Comments on the Quality of English LanguageThe manuscript is generally well-written, but it would benefit from moderate editing to improve clarity and readability. Some sentences are complex and could be simplified for better understanding. Attention to grammar and sentence structure will enhance the overall quality of the text.
These are specific suggestions to improve the English language quality of the document:
· Original phrase: "However, the mechanisms of NAM genes underlying the development and regulation of pseudobulbs remain elusive."
Suggestion: "However, the mechanisms by which NAM genes regulate the development of pseudobulbs remain unclear."
Reason: The original phrase is grammatically correct, but "underlying" can be replaced with a simpler expression for better clarity.
· Original phrase: "The concentration of auxin (IAA) regulated by PIN1 determines the formation and distribution of leaf margin serrations, while the polarity localization of PIN1 is regulated by a class of TFs, NAM/CUC."
Suggestion: "The concentration of auxin (IAA), regulated by PIN1, determines the formation and distribution of leaf margin serrations, while the polarity of PIN1 localization is controlled by a class of TFs, specifically NAM/CUC."
Reason: Small changes in sentence structure can improve flow and clarity.
· Original phrase: "Overall, this study establishes the basis for subsequent research aimed at validating the functional roles of NAM genes in O. hybridum."
Suggestion: "Overall, this study lays the foundation for future research aimed at validating the functional roles of NAM genes in O. hybridum."
Reason: "Subsequent" can be replaced with "future" for simplicity, and "establishes the basis" can be simplified to "lays the foundation."
· Original phrase: "This action promotes ABA biosynthesis and suppresses cytokinin biosynthesis, ultimately delaying the release of tuber dormancy in g.hybridus."
Suggestion: "This action promotes ABA biosynthesis and suppresses cytokinin biosynthesis, ultimately delaying the release of tuber dormancy in G. hybridus."
Reason: "g.hybridus" should be italicized and capitalized as is standard for scientific names.
· Original phrase: "To get deeper insight into the relations between NAM DEGs and endogenous hormones and the roles of NAM gene family in the growth of O. hybridum, we conducted a combined analysis based on both metabolomic and transcriptomic data."
Suggestion: "To gain deeper insights into the relationship between NAM DEGs and endogenous hormones, as well as the role of the NAM gene family in the growth of O. hybridum, we performed a combined analysis using both metabolomic and transcriptomic data."
Reason: Small adjustments to improve grammatical precision, such as using "relationship" instead of "relations" and changing "get" to "gain" for a more formal tone.
Author Response
Dear Reviewer,
We sincerely appreciate your valuable comments and suggestions on our manuscript. Your expertise and guidance are highly valued, and they will undoubtedly help us improve and refine our manuscript. We have carefully revised our manuscript one by one in the revised manuscript (revised sections were in red font) according to your important suggestions, and we hope you could satisfied with our responses and revisions.
In response to your issues:
Comments 1: The introduction provides a good overview of the importance of Oncidium hybridum and the role of the NAM gene family. However, it could benefit from a more detailed review of existing literature, particularly studies that have investigated the regulation of pseudobulb growth in other orchid species. This would help contextualize the study's contributions more clearly.
Response 1: We greatly appreciate your suggestion. After reviewing a substantial amount of relevant literature, we found some studies that have discovered specific genes involved in flower regulation, which also play a significant role in the growth and development of pseudobulbs. We have now included these findings in the introduction, which you can find between lines 45 and 54.
Comments 2: While the methods are generally well described, there are areas where additional detail would enhance reproducibility. For example, more information on the statistical methods used for data analysis would be beneficial. Additionally, it would be helpful to clarify any assumptions made during the bioinformatic analyses.
Response 2: Your suggestion helps enhance the academic value of this paper, and we sincerely thank you for it. A detailed explanation of the data analysis methods and the assumptions of bioinformatics analysis is helpful in accurately reproducing experiments when replicating or expanding the study. We have re-examined the methods section and added the following content:
lines 552 and 585. To enhance the rigor of the academic paper, we have added descriptions regarding the number of biological replicates.
Line 557. To ensure the reproducibility of the experiment, we have added a description of how the RNA sequence data was filtered.
Line564. We clarified that the calculation of FPKM is based on the reads count per gene as determined by FeatureCounts. This enhances the coherence of the method.
Line572. We mentioned the model used for differential statistical analysis.
Line587. We described how the supernatant obtained through centrifugation was diluted.
Line589-596. We provided a detailed description of the specific experimental parameters and solvents used in the LC-MS/MS system.
Comments 3: The results are clearly presented, but some of the figures could be enhanced with more detailed legends to help guide the reader through complex data. For example, in the figures related to gene duplication and phylogenetic trees, additional annotations could improve comprehension.
Response 3: We have added additional figure legends to Figures 2 and Figures 5 as per your suggestion. In the legend for Figure 2, we have explained the meaning of each graphical element and the relationship between the graphics and data in order from the inside out. In the legend for Figure 5, we have clarified the meaning of branch distances in the phylogenetic tree and added classification criteria to help readers better understand the implicit information in the image. You can find the added legends in lines 167-171 and lines 213-215.
Comments 4: The conclusions are well supported by the results. However, the discussion could be expanded to include potential future research directions based on the findings, as well as the broader implications for the field of orchid genetics.
Response 4: Your suggestion prompted us to consider future research directions. Some literature has provided us with ideas, leading us to believe that it would be valuable to further investigate whether these NAM genes have cooperative regulatory functions and to explore their effects on other organs of O. hybridum. We believe that this study has identified target genes for genetic research in Orchidaceae, and in the future, functional validation of these NAM genes could be extended to other Orchidaceae species. We have added the relevant content to the discussion section, which you can find between lines 474 - 488.
Comments 5: The references cited are appropriate and relevant to the research. However, the addition of more recent studies, particularly those published in the last two years, could strengthen the manuscript.
Response 5: Thank you for your suggestion. Without altering the original meaning, we have cited some more recent studies to better illustrate the research background and support discussion. The newly added references are specifically references 2, 3, 41, 42, 43 and 44.
Comments on the Quality of English Language:
The manuscript is generally well-written, but it would benefit from moderate editing to improve clarity and readability. Some sentences are complex and could be simplified for better understanding. Attention to grammar and sentence structure will enhance the overall quality of the text.
These are specific suggestions to improve the English language quality of the document:
Original phrase: "However, the mechanisms of NAM genes underlying the development and regulation of pseudobulbs remain elusive."
Suggestion: "However, the mechanisms by which NAM genes regulate the development of pseudobulbs remain unclear."
Reason: The original phrase is grammatically correct, but "underlying" can be replaced with a simpler expression for better clarity.
Original phrase: "The concentration of auxin (IAA) regulated by PIN1 determines the formation and distribution of leaf margin serrations, while the polarity localization of PIN1 is regulated by a class of TFs, NAM/CUC."
Suggestion: "The concentration of auxin (IAA), regulated by PIN1, determines the formation and distribution of leaf margin serrations, while the polarity of PIN1 localization is controlled by a class of TFs, specifically NAM/CUC."
Reason: Small changes in sentence structure can improve flow and clarity.
Original phrase: "Overall, this study establishes the basis for subsequent research aimed at validating the functional roles of NAM genes in O. hybridum."
Suggestion: "Overall, this study lays the foundation for future research aimed at validating the functional roles of NAM genes in O. hybridum."
Reason: "Subsequent" can be replaced with "future" for simplicity, and "establishes the basis" can be simplified to "lays the foundation."
Original phrase: "This action promotes ABA biosynthesis and suppresses cytokinin biosynthesis, ultimately delaying the release of tuber dormancy in g.hybridus."
Suggestion: "This action promotes ABA biosynthesis and suppresses cytokinin biosynthesis, ultimately delaying the release of tuber dormancy in G. hybridus."
Reason: "g.hybridus" should be italicized and capitalized as is standard for scientific names.
Original phrase: "To get deeper insight into the relations between NAM DEGs and endogenous hormones and the roles of NAM gene family in the growth of O. hybridum, we conducted a combined analysis based on both metabolomic and transcriptomic data."
Suggestion: "To gain deeper insights into the relationship between NAM DEGs and endogenous hormones, as well as the role of the NAM gene family in the growth of O. hybridum, we performed a combined analysis using both metabolomic and transcriptomic data."
Reason: Small adjustments to improve grammatical precision, such as using "relationship" instead of "relations" and changing "get" to "gain" for a more formal tone.
Response 6: We greatly appreciate your detailed suggestions for improving the English language quality of this paper. We have revised the relevant expressions according to your recommendations. Additionally, we have thoroughly reviewed the entire manuscript multiple times and corrected any inappropriate expressions. To further ensure the readability and rigor of this academic article, we have also utilized the professional language editing services provided by MDPI to enhance the quality of the paper. All language modifications are highlighted in red font within the article.
Reviewer 2 Report
Comments and Suggestions for Authors
The manuscript by Liu et al. described the possible role of the NAM family genes in Oncidium hybridum pseudobulb growth by integrating analysis of transcriptome and metabolome. They identified 144 NAM genes in O. hybridum, and analyzed their structural domains, phylogenetics and expression pattern at four stages of pseudobulb development. The results are interesting and provide a foundation for genetic improvement and elucidation of molecular mechanism underlying pseudobulb growth and development. I have several suggestions to improve the manuscript.
1, the qualities of the figures need to be improved;
2, the expression pattern for NAM genes (at least for the expression pattern for those NAM genes expressed in four stages of pseudobulb development) based on RNA-seq data should be confirmed by RT-qPCR assay.
3, it is not clear how many independent biological replicates were conducted for both RNA-seq and metabolite data.
4, some of descriptions are not correct, e.g. 2.6 Expression Patterns of OhNAM Proteins Based on RNA-seq data. RNA-seq data could provide information of stable mRNA, but no information for expression of protein (the expression of mRNA and protein is not correlated in most of cases).
5, for the title, “Integrative analysis of transcriptome and metabolome reveals the pivotal role of the NAM gene family in Oncidium hybridum pseudobulb growth”, “the NAM gene family” should be “the NAM family genes”?
Comments on the Quality of English LanguageThe writing should be smoothed.
Author Response
Dear Reviewer,
We sincerely appreciate your valuable comments and suggestions on our manuscript. Your expertise and guidance are highly valued, and they will undoubtedly help us improve and refine our manuscript. We have carefully revised our manuscript one by one in the revised manuscript (revised sections were in red font) according to your important suggestions, and we hope you could satisfied with our responses and revisions.
In response to your issues:
Comments 1: the qualities of the figures need to be improved
Response 1: Thank you for your suggestion. We have made adjustments to some images without altering the information they convey. For example, we added a scale bar to Figure 12, which makes the size of the experimental samples clearer and highlights the differences in pseudobulbs at different growth stages. Additionally, we corrected font errors in some images, such as italicizing the gene names in Figures 1 and 2.
Comments 2: the expression pattern for NAM genes (at least for the expression pattern for those NAM genes expressed in four stages of pseudobulb development) based on RNA-seq data should be confirmed by RT-qPCR assay.
Response 2: Thank you for your suggestion. We selected three crucial genes related to pseudobulb development and conducted qPCR analysis. The methods and results can be found in sections 4.8 (lines 617-627) and 2.8 (lines 342-351), respectively.
Comments 3: it is not clear how many independent biological replicates were conducted for both RNA-seq and metabolite data.
Response 3: Thank you for your reminder. Clarifying the number of independent biological replicates in the experiments helps enhance the rigor of the academic paper. We have added description regarding the number of biological replicates in sections 4.5 and 4.6, which you can find at lines 552 and 585, respectively.
Comments 4: some of descriptions are not correct, e.g. 2.6 Expression Patterns of OhNAM Proteins Based on RNA-seq data. RNA-seq data could provide information of stable mRNA, but no information for expression of protein (the expression of mRNA and protein is not correlated in most of cases)
Response 4: We have revised the title of section 2.6 to “Expression Patterns of OhNAM Genes Based on RNA-seq Data” as per your suggestion, which you can find at line 279.
Comments 5: For the title, “Integrative analysis of transcriptome and metabolome reveals the pivotal role of the NAM gene family in Oncidium hybridum pseudobulb growth”, “the NAM gene family” should be “the NAM family genes”?
Response 5: Thank you for your suggestion. We have revised the title to “Integrative analysis of transcriptome and metabolome reveals the pivotal role of the NAM family genes in Oncidium hybridum Lodd. pseudobulb growth”. After reviewing a substantial amount of literature, we found that both “gene family” and “family genes” are used in the literature. We discussed that “The NAM gene family” emphasizes NAM genes as a unit or category, while “The NAM family genes” highlights specific genes within the NAM family. In our study, we identified 11 NAM genes that regulate JA and ABA through the integrative analysis of transcriptome and metabolome. “The NAM family genes” better conveys our focus.
Comments on the Quality of English Language: The writing should be smoothed.
Response 6: We greatly appreciate your advice for improving the English language quality of this paper. We have thoroughly reviewed the entire manuscript multiple times and corrected any inappropriate expressions. To further ensure the readability and rigor of this academic article, we have also utilized the professional language editing services provided by MDPI to enhance the quality of the paper. All language modifications are highlighted in red font within the article.
Reviewer 3 Report
Comments and Suggestions for Authors
In this study, 144 members of the NAM gene family were identified using Oncidium hybridum as the material. The characteristics of these members were elucidated in terms of physicochemical properties, chromosomal localization, gene duplication, gene structure analysis, and phylogenetic relationships. Subsequently, differences in NAM gene expression during developmental stages were analyzed from the perspectives of transcriptomics and metabolomics, providing new insights into the molecular mechanisms of pseudobulb growth and development in O. hybridum. However, there are the following shortcomings that could be further improved:
1. The gene names in Figure 1 and Figure 2 should be italicized. There are also errors in the main text where gene names are not italicized; please carefully check for these in the subsequent revisions.
2. The name of the species should include the author citation when mentioned for the first time.
3. For gene family members that are uniquely located in both chloroplasts and nuclei, multiple subcellular localization prediction tools could be used for verification to make the results more convincing.
4. The study only analyzed collinearity within the species; collinearity compared with Arabidopsis thaliana could be added to the text.
5. A size scale should be added to Figure 10.
Comments on the Quality of English LanguageModerate editing of English language required.
Author Response
Dear Reviewer,
We sincerely appreciate your valuable comments and suggestions on our manuscript. Your expertise and guidance are highly valued, and they will undoubtedly help us improve and refine our manuscript. We have carefully revised our manuscript one by one in the revised manuscript (revised sections were in red font) according to your important suggestions, and we hope you could satisfied with our responses and revisions.
In response to your issues:
Comments 1: The gene names in Figure 1 and Figure 2 should be italicized. There are also errors in the main text where gene names are not italicized; please carefully check for these in the subsequent revisions.
Response 1: Thank you for your reminder. You mentioned that Figure 1, Figure 2, and some gene names in the article should be italicized. We have made the necessary changes to the corresponding images and text. The specific revisions can be found in lines 172 (Table 1), line 321, line 335 (Table2), and line 438 of the article.
Comments 2: The name of the species should include the author citation when mentioned for the first time.
Response 2: Thank you for your reminder. We reviewed the entire manuscript and added the author citation at the first mention of each species name in the article. The specific revisions are as follows:
Line 3, Line 12, and line 39 “Oncidium hybridum Lodd.”
Line 133 and line 157 “Arabidopsis thaliana (L.) Heynh.”
Line 91 “Chrysanthemum lavandulifolium (Fisch. ex Trautv.) Makino”
Line 97 “Gladiolus hybridus Hort.”
Line 401 “Lagerstroemia indica L.”
Line 454 “Cucurbita pepo L.”
Comments 3: For gene family members that are uniquely located in both chloroplasts and nuclei, multiple subcellular localization prediction tools could be used for verification to make the results more convincing.
Response 3: You mentioned that various subcellular localization prediction tools could be used to validate the gene family members localized to both chloroplasts and nuclei. We greatly appreciate your suggestion and have used Plant-mPLoc, a tool specifically designed for predicting the subcellular localization of plant proteins, to perform subcellular localization predictions for all NAM gene family members. The results indicate that the subcellular localization predictions obtained using Plant-mPLoc are consistent with those obtained using WoLF PSORT, suggesting that the results are reliable. We have revised sections 2.1 and 4.1 accordingly, as shown in lines 129-132 and 506-509.
Comments 4: The study only analyzed collinearity within the species; collinearity compared with Arabidopsis thaliana could be added to the text.
Response 4: We appreciate your suggestion to include a collinearity analysis between the two species. We reviewed a large number of studies and found that some research on gene families includes collinearity analyses between multiple species. This indeed helps determine whether the functions of these genes are conserved across different species and aids in understanding the evolutionary history of the genome and its impact on the function and structure of gene families. We have added a collinearity analysis between the two species in the manuscript. You can find the relevant results, discussion and method in Section 2.2 (lines 156-160), Section 3 (line 390-394) and Section 4.2 (lines 522-525).
Comments 5: A size scale should be added to Figure 10.
Response 5: Thank you for your suggestions. We have added a size scale to Figure 10, which indeed makes the dimensions of the experimental samples clearer and also highlights the differences at various growth stages of the pseudobulb. As we have added two figures, the revised Figure 10 has been renamed to Figure 12 and is now located on line 548 of Section 4.4.
Comments on the Quality of English Language: Moderate editing of English language required.
Response 6: We greatly appreciate your advice for improving the English language quality of this paper. We have thoroughly reviewed the entire manuscript multiple times and corrected any inappropriate expressions. To further ensure the readability and rigor of this academic article, we have also utilized the professional language editing services provided by MDPI to enhance the quality of the paper. All language modifications are highlighted in red font within the article
Round 2
Reviewer 3 Report
Comments and Suggestions for Authors
The manuscript has been improved greatly after the revision and can be accepted now.
Comments on the Quality of English LanguageThe English writing is good and only minor editing of English language is required.